# *In-Vitro* Antioxidant, Hypoglycemic Activity, and Identification of Bioactive Compounds in Phenol-Rich Extract from the Marine Red Algae *Gracilaria edulis* (Gmelin) Silva

**DOI:** 10.3390/molecules24203708

**Published:** 2019-10-15

**Authors:** Thilina L. Gunathilaka, Kalpa W. Samarakoon, P. Ranasinghe, L. Dinithi C. Peiris

**Affiliations:** 1Department of Zoology, Faculty of Applied Sciences (Center for Instrumentation Facility & Center for Biotechnology), University of Sri Jayewardenepura, Nugegoda 10250, Sri Lanka; gunathilakathilina2@gmail.com; 2National Science and Technology Commission, Dudley Senanayake Mawatha, Colombo 8 00800, Sri Lanka; kalpa.samarakoon@gmail.com; 3Industrial Technology Institute, Halbarawa Gardens, Malabe 10115, Sri Lanka; pathmasiri@iti.lk

**Keywords:** *Gracillaria edulis*, methanol extract, fractionation, α-amylase, α-glucosidase, antiglycation, glucose diffusion

## Abstract

Obesity and diabetes are major metabolic disorders which are prevalent worldwide. Algae has played an important role in managing these disorders. In this study, *Gracilaria edulis*, a marine red algae, was investigated for antioxidant and hypoglycemic potential using in vitro models. De-polysaccharide methanol extract of *G. edulis* was sequentially partitioned with hexane, chloroform, ethyl acetate, and antioxidants, and hypoglycemic potentials were evaluated using multiple methods. High antioxidant potential was observed in the ethyl acetate fraction in terms of ferric reducing antioxidant power, iron chelating, and DPPH and ABTS radical scavenging activities, while the crude methanol extract exhibited potent oxygen radical-absorbance capacity. Potent α-amylase inhibitory activity was observed in the ethyl acetate fraction, while the ethyl acetate fraction was effective against α-glucosidase inhibition. Glucose diffusion was inhibited by the ethyl acetate fraction at 180 min, and the highest antiglycation activity was observed in both chloroform and ethyl acetate fractions. Additionally, gas chromatography-mass spectrometry analysis of the ethyl acetate fraction revealed the presence of several potent anti-diabetic compounds. In conclusion, *G. edulis* exhibited promising antidiabetic potential via multiple mechanisms. The ethyl acetate fraction exhibited the strongest hypoglycemic and antiglycation potential among the four fractions, and hence the isolation of active compounds is required to develop leads for new drugs to treat diabetes.

## 1. Introduction

The incidences of type 2 diabetes and obesity have increased globally due to rapid urbanization and unhealthy diets. More than 90% of patients with diabetes mellitus are either overweight or obese [1]. Diabetes mellitus is a chronic disorder that is linked with persistent hyperglycemia due to the deficiency of insulin secretion. The World Health Organization (WHO) has estimated that by 2035, the incidence of diabetes mellitus and impaired glucose tolerance will increase by up to 592 million and 471 million people, respectively [2]. Type 1 diabetes is widespread among Northern European countries, while type 2 diabetes is most common in African and South Asian countries. For instance, type 2 diabetes is prevalent among the Sri Lankan population [3]. According to recent statistics, one in every five Sri Lankan adults either suffers from diabetes or is in the prediabetes stage [4]. The consumption of more refined fast-release staple carbohydrate food is considered as the major cause for the progression of obesity. Carbohydrate-rich diets release glucose quickly into the bloodstream, thus increasing the levels of blood sugar and insulin [5]. High blood glucose level is linked with increased risk of hypertension, retinopathy, nephropathy, neuropathy, and macrovascular diseases. These health complications result in an increased risk of morbidity and mortality, and hence reduce the life expectancy of diabetic patients [4].

The inhibition of carbohydrate digestive enzymes—α-amylase and α-glucosidase—is one of the significant alternatives to the management of chronic hyperglycemia in diabetic patients [1]. Polyphenols purified from plants are good inhibitors of vital enzymes responsible for carbohydrate digestion. The enzymes alpha amylase and α-glucosidase are involved in carbohydrate metabolism, and act synergistically to digest starch [6]. Alpha amylase hydrolyzes the alpha bonds present in insoluble starch molecules, while α-glucosidase catalyzes the final step of carbohydrate digestion to convert disaccharides into glucose. The inhibition of such enzymes leads to a reduction of starch breakdown and an increase in postprandial blood glucose level; thus, enzyme inhibitors can be used as therapeutic agents for the development of novel drugs to treat diabetes [6].

Recent investigations have reported an association between obesity and chronic inflammation in adipose tissues. As a result of obesity, the amount of adipose tissue tends to increase and the tissue undergoes molecular and cellular alterations. Adipocytokines secreted by the adipocytes in adipose tissues can induce production of reactive oxygen species, thus leading to oxidative stress [7]. Oxidative stress is tightly linked with the pathophysiological process of chronic inflammatory conditions such as diabetes mellitus [8]. Therefore, researchers have primarily focused on natural products to discover novel preventive and regenerative therapies to combat oxidative stress and postprandial hyperglycemia with minimum side effects.

Marine seaweeds rich in bioactive metabolites play a significant role in the development of novel drugs and nutraceuticals. Due to the bioactive compounds they contain—namely polyphenols, sterols, alkaloids, flavonoids, tannins, proteins, essential fatty acids, enzymes, vitamins, and carotenoids—marine seaweeds are able to withstand harsh environments [9]. *Gracilaria edulis* (Gmelin) Silva is a red algae belonging to the family Gracilariaceae, and it has attracted widespread attention due to its biological and pharmacological properties and various therapeutic benefits such as antidiabetic, antioxidant, antimicrobial, anticoagulant, anti-inflammatory, and antiproliferative activities [10]. Shanura et al. [11] established the anti-inflammatory activity of methanol extract and fractions of *G. edulis* against lipopolysaccharide-induced inflammatory responses, while Koneri and Jha [12] documented the antidiabetic potential of methanol extracts of *G. edulis* against fructose-induced type 2 diabetes mellitus in male rats. Patra and Muthuraman [13] revealed the anticancer activity of ethanol extracts of *G. edulis* against ascites tumors in mice. Although past studies focused on several biological properties of *G. edulis,* this is the first study carried out in Sri Lanka to investigate the antioxidant and hypoglycemic activities of *G. edulis* using multiple in vitro mechanisms. The present study aimed to appraise the antidiabetic potential of *G. edulis* through inhibitory activities of carbohydrate digestive enzymes, glucose diffusion, and protein glycation. We also attempted to identify the bioactive compounds present in *G. edulis* that are responsible for the above pharmacological activities.

## 2. Results

### 2.1. Quantification of Total Phenolic, Flavonoid, and Alkaloid Contents of G. edulis

Results obtained for the crude methanol extract and four fractions of *G. edulis* with increasing polarity were used to evaluate the total phenolic, flavonoid, and alkaloid contents of *G. edulis.* These results are shown in Table 1. The alkaloid content in the crude methanol extract and in the hexane and chloroform fractions was higher than the phenol and flavonoid contents, whereas the phenol content in the ethyl acetate and aqueous fractions was higher than the flavonoid and alkaloid contents.

The total phenolic content in the ethyl acetate fraction (2414.51 ± 50.34 µg GAE/g) was higher than that in the crude methanol extract and the hexane, chloroform, and aqueous fractions. The lowest total phenolic content was observed in the chloroform fraction. Similarly, a significant difference was observed in the phenolic content of the crude methanol extract and the four fractions (*p* < 0.05). The total flavonoid content of the crude methanol extract and all fractions increased in the order chloroform fraction < crude methanol extract < hexane fraction < aqueous fraction < ethyl acetate fraction, with respective contents of 289.39 ± 9.55, 541.02 ± 51.84, 688.60 ± 9.55, 786.95 ± 62.04, and 1461.49 ± 75.22 µg QE/g. Among the crude methanol extract and four fractions, total alkaloid content decreased in the order crude methanol extract > chloroform fraction > hexane fraction > ethyl acetate fraction > aqueous fraction, with respective values of 7177.72 ± 63.04, 2875.54 ± 22.29, 1656.97 ± 45.80, 1073.75 ± 45.88, and 522.34 ± 67.13 µg PE/g. 

### 2.2. In Vitro Antioxidant Activity

The antioxidant capacity of crude methanol extract and fractions of *G. edulis* were determined using 1,1-diphenyl-2-picrylhydrazine (DPPH) and 2,2′-azino-bis (3-ethylbenzothiazoline-6-sulphonic acid) (ABTS) radical scavenging activities, ferrous ion chelating assay, ferric reducing antioxidant power, and oxygen radical-absorbance capacity. The results obtained for the antioxidant capacity are presented in Table 2.

The highest DPPH radical scavenging activity was observed in the ethyl acetate fraction (IC_50_: 3.17 ± 0.04 mg/mL) and the crude methanol extract (IC_50_: 3.19 ± 0.02 mg/mL). By comparison, the standard Trolox had free radical scavenging activity of IC_50_: 0.011 mg/mL. In this study, the reduction of DPPH occurred in a concentration-dependent manner, as observed from the high reduction of DPPH (higher radical activity) at 3.75 mg/mL concentrations (Figure A1). The most potent ABTS radical scavenging activity was observed in the ethyl acetate fraction (IC_50_: 0.41 ± 0.02 mg/mL), while the lowest activity was observed in the crude methanol extract (IC_50_: 0.56 ± 0.01 mg/mL). Moreover, a significant positive correlation was observed between the radical scavenging activity of DPPH (r = 0.91) and that of ABTS (r = 0.85) with the total phenol content of the ethyl acetate fraction. The highest ferrous iron chelating activity (FICA) was observed in the ethyl acetate fraction (IC_50_: 2.22 ± 0.01 mg/mL), while the lowest ferrous ion chelating activity was observed in the crude methanol extract (IC_50_: 9.23 ± 0.19 mg/mL); the standard EDTA (ethylenediaminetetraacetic acid) exhibited a chelating activity of IC_50_: 0.019 mg/mL. Ferric reducing antioxidant powder (FRAP) and oxygen radical absorbance capacity (ORAC) are the most frequent measures used to determine antioxidant activity, which is expressed as Trolox equivalent antioxidant capacity. The ethyl acetate fraction of *G. edulis* showed the highest reducing ability (8.51 ± 0.09 mg TE/g), while crude methanol extract (1.61 ± 0.19 mg TE/g) showed potent oxygen radical absorbance capacity.

### 2.3. In Vitro α-Amylase Inhibitory Assay

The α-amylase inhibitory potential of *G. edulis* methanol extract and four fractions was evaluated using starch as the substrate and acarbose as the positive control. Acarbose, crude methanol extract, and the four fractions exhibited a dose-dependent enzyme inhibition (Figure A2).

The ethyl acetate fraction (IC_50_: 279.48 ± 5.62 µg/mL) of *G. edulis* exhibited more potent α-amylase inhibitory activity, whereas the lowest activity was observed in the hexane fraction (IC_50_: 393.04 ± 4.73 µg/mL) (Table 3) compared to the standard acarbose (IC_50_: 87.43 µg/mL). The inhibition of the α-amylase enzyme exhibited by the ethyl acetate fraction varied from 10% at 12.5 µg/mL to 64% at 400 µg/mL assay concentration.

### 2.4. In Vitro α-Glucosidase Inhibitory Assay

All the fractions and the crude methanol extract of *G. edulis* exhibited a dose-dependent inhibition of the α-glucosidase enzyme with different degrees of potential (Figure A3). The inhibitory activity on *α*-glucosidase enzymes of the crude methanol extract and the fractions of *G. edulis* are presented in Table 3. The ethyl acetate fraction, which exhibited the lowest IC_50_ of 87.92 ± 1.62 µg/mL, was considered to be a more potent *α*-glucosidase inhibitor than the crude methanol extract and the other three fractions. The inhibition of α-glucosidase exhibited by the ethyl acetate fraction varied from 6% (4.16 µg/mL) to 68% (133.3 µg/mL). The hexane fraction, which had the highest IC_50_ of 163.90 ± 5.23 µg/mL, exhibited the weakest *α*-glucosidase inhibitory activity.

### 2.5. Glucose Diffusion Inhibitory Activity

The inhibitory activity of glucose diffusion was determined using a dialysis tube containing the sample and glucose that had been soaked in NaCl solution. The diffusion of glucose into the external solution was measured by glucose oxidase kit every 30 min for 3 h. The effects of the methanol extract and the four fractions of *G. edulis* (1000 µg/mL) on glucose diffusion are represented in Figure 1. The inhibitory activity of acarbose or the reference drug was considered as 100%, while the glucose diffusion of the control at 180 min was considered as 100% with a glucose concentration of 57.65 ± 1.67 µg/mL in the external solution, compared to the acarbose standard (22.79 ± 0.47 µg/mL). Among the tested fractions, the ethyl acetate fractions exhibited the maximum inhibition of glucose diffusion at 180 min, and the glucose concentration of the external solution was found to be 38.15 ± 1.11 µg/mL. The inhibition of glucose diffusion by the hexane fraction (52.01 ± 0.96 µg/mL) and aqueous fraction (52.69 ± 1.31 µg/mL) were similar compared to the glucose concentration in the external solution (Table A1).

### 2.6. Antiglycation Activity

The anti-glycation activity of the crude methanol extract and fractions of *G. edulis* was determined using bovine serum albumin as a protein source. The crude methanol extract and all four fractions showed dose-dependent antiglycation activity (Figure 2). As shown in Table 3, among the tested fractions, the most potent antiglycation activity was observed with the chloroform fraction (IC_50_: 258.23 ± 3.24 µg/mL), followed by the ethyl acetate fraction (IC_50_: 586.54 ± 4.37 µg/mL), compared to the standard rutin (IC_50_: 11.55 ± 0.82 μg/mL). The aqueous fraction, which had the highest IC_50_ value of 723.78 ± 12.81 µg/mL, exhibited the weakest antiglycation activity.

### 2.7. GC-MS or Gas Chromatography-Mass Spectrometry Analysis of Extract and Solvent Fractions

The ethyl acetate fraction of *G. edulis*, which showed promising biological activities, was subjected to gas chromatography-mass spectrometry (GC-MS) analysis. The chromatogram obtained for the ethyl acetate fraction is presented in Figure 3.

Based on the retention time and molecular weights of the GC-MS chromatogram, six compounds were identified in the ethyl acetate fraction of *G. edulis*. As listed in Table 4, these compounds included 2,5-dimethylhexane-2,5-dihydroperoxide, phthalic acid-6-ethyloct-3-yl 2-ethylhexyl ester, 1H-Indole-2-carboxylic acid, 6-(4-ethoxyphenyl)-3-methyl-4-oxo-4,5,6,7-tetrahydro-isopropylester, 2,3,5-Trichlorobenzaldehyde, Benz(b)1,4-oxazepine-4 (5H)-thione, 2,3-dihydro-2,8-dimethyl, and 2-acetoxymethyl-3-(methoxycarbonyl) biphenylene.

Among these active compounds, 1*H*-Indole-2-carboxylic acid,6-(4-ethoxyphenyl)-3-methyl-4-oxo-4,5,6,7-tetrahydro-isopropyl ester was identified as possessing antidiabetic activity through insulin sensitizing and glucose lowering effects [17]. In addition, most of the indole derivatives act as an activators of glycogen synthase enzyme, which involved in the glycogen synthesis pathway [18], while 2,5-dimethylhexane-2,5-dihydroperoxide has been identified to have antioxidant properties which help to prevent the development of oxidative stress related to diabetes [14,15].

## 3. Discussion

Recently, red marine algae has received significant attention due to its immense therapeutic benefits. However, only a limited number of studies have been performed on Sri Lankan marine algae. Therefore, to discover the therapeutic potential of tropical marine algae against diabetes and obesity, we studied *G. edulis,* a red algae from the northwestern coast of Sri Lanka to investigate its mechanisms of action.

Polyphenols, flavonoids, and alkaloids are secondary plant metabolites that have shown therapeutic benefits and are considered as potential sources of antioxidants. The polyphenol, flavonoid, and alkaloid contents of plants may vary depending on environmental factors, soil type, sun exposure, rainfall, etc. [22]. As determined by the present study, the ethyl acetate fraction of *G. edulis* contained the highest phenol and flavonoid content, whereas the highest alkaloid content was observed in the crude methanol extract of *G. edulis*. The present study found comparatively lower phenolic content compared to the results of Ganesan et al. [23], who reported 3.98 mg/mL phenolic content. This variation may be due to the different sample collection locations, temperature conditions, and stress tolerance [24].

Oxidative stress is linked with the development of diabetes, and increases in accumulated fat in obese individuals through the activation of nicotinamide adenine dinucleotide phosphate (NADPH) oxidase and the impaired production of adipocytokines [25]. The antioxidant activity determined by the DPPH assay revealed high free radical scavenging activities in the ethyl acetate fraction and crude methanol extract of *G. edulis*, which occurred in a concentration-dependent manner (Figure A1). In the present study, strong DPPH free-radical scavenging ability was observed at 950 µg/mL of the ethyl acetate fraction (21.06%) and crude methanol extract (19.52%). This result contradicts a previous report in India [23]; compared to the present study, [16] reported lower DPPH free radical scavenging activity of the ethyl acetate fraction (4.73%) and the crude methanol extract (5.20%). The higher antioxidant activity observed in the present study may be due to the use of different extraction procedures and the differences in the phenolic compounds responsible for antioxidant activity.

The highest ABTS radical scavenging activity was observed in the ethyl acetate fraction of *G. edulis,* with a significant positive correlation being observed with total phenol content (r = 85). The reducing power of the sample depends on the available phenol and flavonoid contents of the sample [26]. In the present study, the ethyl acetate fraction exhibited the highest FRAP activity, which can be attributed to the presence of phenolic or flavonoid compounds with functional groups such as hydroxyl and carbonyl, which leads to the reduction or inhibition of oxidation. In contrast, Francavilla et al. reported the highest ABTS radical scavenging activity and ferric reducing antioxidant power in ethyl acetate fraction (0.43 and 0.809 mmol TE/g) of *G.edulis* collected seasonally in the Lesina Lagoon in Italy during the period of July [27]. Similarly, the ethyl acetate fraction of the present study exhibited higher reducing power compared to the previous study, which was collected in February. Therefore, differences in reducing power might be due to the seasonal variation.

The chelating power of the sample was determined using ferrozine reagent. Ferrozine can chelate with Fe^2+^, forming a red-colored ferrozine-Fe^2+^ complex [28]. According to the chelation activity, the highest FICA was observed in the ethyl acetate fraction. The ORAC assay determines the oxidative degradation of fluorescein in the presence of a free radical generator, such as an azo compound, for example, 2-azobis (2-amidinopropane) dihydrochloride [29]. The ORAC assay confirmed the high antioxidant capacity of the crude methanol extract of *G. edulis*. The high phenol and flavonoid contents of the ethyl acetate fraction are responsible for the potent antioxidant activity of *G. edulis*. Additionally, the presence of bioactive compounds such as 2,5-dimethylhexane-2,5-dihydroperoxide may also contribute to the antioxidant activity of *G. edulis*.

The inhibition of key metabolic carbohydrate-digesting enzymes is one of the main strategies to determine the antidiabetic activity of medicinal plants [30]. Therefore, natural bioactive compounds that reduce blood glucose levels by inhibiting the key metabolic enzymes (α-amylase, α-glucosidase) and glucose absorption can be considered to be useful for the management of diabetes [31]. From the results, it is evident that the ethyl acetate fraction of *G. edulis* showed potential inhibitory activities with an effective dose for inhibition of α-amylase (IC_50_: 279.48 ± 5.62 µg/mL) and α-glucosidase (IC_50_: 87.92 ± 1.62 µg/mL) enzymes that are comparable to the standard drug acarbose (Table 3). The potential enzyme-inhibiting activity of the ethyl acetate fraction of G*. edulis* can be attributed to the presence of phytochemicals such as phenolic and flavonoid compounds. Senthil and Sudha [32] reported potent effective doses of inhibitory activity of α-amylase (IC_50_: 83 µg/mL) and α-glucosidase (IC_50_: 46 µg/mL) in an aqueous extract of *G. edulis* collected from India. The difference between the results obtained in the present study may be due to the different solvents used for the extraction method; in the present study, polyphenol was initially used to extract methanol, while the study conducted by Senthil and Sudha used water as a solvent.

The glucose diffusion inhibition test was carried out to evaluate the effect of methanol extract and fractions of *G.edulis,* with respect to its glucose retardation activity across the dialysis tube. The glucose entrapment ability of the crude methanol extract and four fractions were found to be significantly different at different times. Among them, the ethyl acetate fraction of *G.edulis* exhibited a significant glucose entrapment ability, which decreased the glucose movement into the external solution at 180 min compared to the control. The fact that the ethyl acetate fraction exhibited the highest inhibition of glucose diffusion may be due to the presence of insoluble fiber particles which entrap glucose molecules [33,34]. The dialysis tube method is a simple technique, which only determines the potential effect of methanol extract and fractions of *G.edulis* to retard the glucose diffusion through the normal dialysis membrane, whereas in the intestinal tract, transportation of glucose is assisted by glucose transporters incorporated with other molecules, in addition to the intestinal contractions [35]. Therefore, further in vivo studies should be carried out to determine the real effect of methanol extract and fractions of *G. edulis* on glucose diffusion.

Chronic hyperglycemia in diabetic patients leads to the progression of microvascular and macrovascular complications. The high level of blood glucose leads to the formation of a complex between glucose and plasma proteins through non-enzymatic reactions and forms AGEs, which are associated with the pathogenesis of vascular complications in diabetes, renal failure, Alzheimer’s disease, aging, and other chronic diseases [27]. The present study is the first to report the inhibitory activity of the methanol extract and four fractions of *G. edulis* on the formation of AGE products. The results of the study show that the lowest effective dose of inhibition of AGE formation was exhibited by chloroform (IC_50_:258.23 ± 3.24 µg/mL) and ethyl acetate fractions (IC_50_: 586.54 ± 4.37 µg/mL) of *G. edulis,* compared to the standard drug rutin. The antiglycation activity of the chloroform and ethyl acetate fractions may be due to the presence of phenolic and flavonoid compounds, which are significantly correlated with anti-glycation activity [36].

Furthermore, GC-MS analysis of the ethyl acetate fraction of *G. edulis* revealed the presence of active compounds with strong antioxidant and antidiabetic activity, including 1*H*-Indole-2-carboxylic acid,6-(4-ethoxyphenyl)-3-methyl-4-oxo-4,5,6,7-tetrahydro-isopropyl ester and 2,5-dimethylhexane-2,5-dihydroperoxide. 1*H*-Indole-2-carboxylicacid,6-(4-ethoxyphenyl)-3-methyl-4-oxo-4,5,6,7 tetrahydro-isopropyl ester is an indole derivative, which has insulin sensitizing and glucose lowering effects [17]. In addition, most of the indole derivatives act as an activator of the glycogen synthase enzyme, which is involved in the glycogen synthesis pathway [18], while 2,5-dimethylhexane-2,5-dihydroperoxide was identified to have antioxidant properties, which help to prevent the development of oxidative stress related to diabetes [14,15].

## 4. Materials and Methods

### 4.1. Chemicals and Reagents

Trolox, DPPH, trichloroacetic acid, ABTS, potassium persulpahte, 2,4,6-tripyridyl-s-triazine (TPTZ), soluble starch, Folin–Ciocalteu reagent, alpha glucosidase from *Saccharomyces cerevisiae*, gallic acid, acarbose, quercetin, aluminum chloride, *p*-nitrophenyl a-d-glucopyranoside, bovine serum albumin (BSA), and alpha-amylase were purchased from Sigma Aldrich (Allentown, PA, USA). The chemicals and reagents used for all experiments were of analytical grade.

### 4.2. Collection of Algae Sample

Permission to collect an algae sample was obtained from the Department of Wildlife Conservation (permit number WL/3/280/17)**.** Marine red algae (*Gracillaria edulis*) was manually collected from Kalpitiya, Sri Lanka (6^0^ 4′ 54.19” N; 80^0^ 8′ 51.78” E). The samples were identified based on their morphological characteristics by Dr. Kalpa Samarakoon. The collected samples were cleaned and washed with fresh water to remove salt, sand, and other debris. The samples were freeze-dried using a LyoBeta freeze drying unit (Telstar), powdered, and stored at −20 °C until further use.

### 4.3. Preparation of Gracillaria edulis Extract and Its Solvent Fractions

The extraction procedure was conducted according to the method of Lakmal et al., [37] with some modifications. Homogenized *G. edulis* powder was extracted three times using 70% methanol, and was then subjected to sonication (Clifton, 91695) at 25 °C. Polyphenols were separated using 70% ethanol (*v*/*w*% = 1:25) and allowed to stand overnight. The supernatant was separated by centrifugation (Centurion K241R, Surrey, UK) at 12,000 rpm before being filtered to separate the polyphenol portion. A portion of the de-polysaccharide methanol extract was subjected to sequential solvent–solvent partition with hexane, chloroform, and ethyl acetate, respectively. Finally, the four fractions and crude methanol extract were dried under vacuum (BUCHI, Rotavapor, R-300, New Castle, DE, USA) and used to conduct the assays.

### 4.4. Quantification of Phenol, Flavonoid, and Alkaloid Contents

The total phenol content of the methanol extract and fractions was evaluated using the Folin–Ciocalteu method (*n* = 4). Different concentrations (5, 10, and 20 mg/mL) of methanol extract and the four fractions were prepared by diluting with distilled water. The sample (20 µL), diluted Folin–Ciocalteu reagent (110 µl), and 10% sodium carbonate solution (70 µL) were mixed and incubated at room temperature for 30 min. Absorbance was measured at 765 nm. Gallic acid was used as the standard, and results were expressed as mg gallic acid equivalent per 1 g of dry weight of the extract/fraction [38].

The total flavonoid content of the methanol extract and fractions (*n* = 4) of *G. edulis* were evaluated by the aluminium chloride method [39]. Different concentrations (5, 10, and 20 mg/mL) of the methanol extract and fractions were prepared by diluting with methanol. The absorbance before adding the 2% aluminum chloride was taken at 415 nm. A total of 100 µl of 2% aluminium chloride solution was added and incubated at room temperature for 10 min. A plate reading was recorded at 415 nm. Quercetin was used as a standard, and results was expressed as mg quercetin equivalent per 1 g of dry weight of the extract or reaction.

The total alkaloid content of the methanol extract and fractions (*n* = 4) of *G. edulis* was evaluated by the method described by Sreevidya et al. [40] with some modifications. Different concentrations (5, 10, and 20 mg/mL) of the methanol extract and fractions of *G. edulis* were prepared by diluting with 95% ethanol and pH was adjusted to 2–2.5. A total of 100 µL of sample was mixed with 200 µL of Dragendorff reagent and centrifuged at 5000 rpm for 5 min. The precipitate was separated and washed with 95% ethanol. A total of 200 µL of 1% disodium sulfide solution was added, and the resulting brownish-black precipitate was centrifuged at 5000 rpm for 5 min. The supernatant was discarded and the pellet was dissolved in concentrated HNO_3_ and diluted up to 1 mL with distilled water. A total of 100 µL of the solution was pipetted and mixed with 500 µL of 3% thiourea. Absorbance was recorded at 460 nm. Piperine was used as the standard, and the results were expressed as mg piperine equivalent per 1 g of dry weight of extract or fraction.

### 4.5. In Vitro Antioxidant Activity

#### 4.5.1. DPPH Radical Scavenging Activity

The DPPH scavenging activity of the extracts and fractions of *G. edulis* (*n* = 4) was determined using the method described by Blois [41]. A concentration series (3.75, 1.875, 0.938, 0.469, 0.234, 0.1170, and 0.058 mg/mL) of algal samples was prepared in methanol. A total of 50 µL of sample was mixed with methanol and absorbance before adding the DPPH solution was taken at 517 nm. Subsequently, DPPH solution was added and incubated in a dark area for 15 min at 25 °C. After incubation, a plate reading was taken at 517 nm. Trolox was used as the standard, and the DPPH scavenging activity was expressed as mg Trolox equivalent per 1 g of dry weight of extract or fraction.

#### 4.5.2. ABTS^+^ Radical Scavenging Activity

The ABTS^+^ scavenging activity was measured using the method described by Re et al. [42]. Before conducting the experiment, ABTS^+^ radical was produced by incubating ABTS tablet (10 mg) with 2.5 mM potassium persulpahte solution (2.5 mL). Five different concentrations (0.5, 0.25, 0.125, 0.063, and 0.031mg/mL) of the methanol extract and fractions (*n* = 4) were diluted in 50 mM PBS (pH 7.4) and absorbance before adding ABTS reagent was taken at 734 nm. A total of 40 µL of diluted ABTS+ reagent was added and incubated for 10 min at room temperature. After incubation, a plate recording was taken at 734 nm. Trolox was used as the standard antioxidant and the results were expressed as mg Trolox equivalent per 1 g of dry weight of extract or fraction.

#### 4.5.3. Ferric Reducing Antioxidant Power (FRAP)

Ferric reducing antioxidant power was measured according to the method of Benzie and Szeto [43], with some modifications. Different concentrations (5, 2.5, 1, and 0.5 mg/mL) of the methanol extract and four fractions (*n* = 4) were diluted in 300 mM acetate butter (pH 3.6). Before conducting the experiment, FRAP reagent was prepared using 300 mM acetate buffer (pH 3.6), 10 mM of 2,4,6-tripyridyl-s-triazine (TPTZ), and 20 mM FeCl_3_ solution mixed with a ratio of 10:1:1 and incubated at 37 °C for 10 min. The FRAP reagent (150 µL), acetate buffer (30 µL), and algae samples (20 µL) were mixed together and incubated for 8 min at room temperature. After incubation, a plate reading was made at 600 nm. Trolox was used as the standard and the results were expressed as mg Trolox equivalent per 1 g dry weight of extract or fraction.

#### 4.5.4. Ferrous Iron Chelating Capacity (FICC)

The ferrous iron chelating capacity (FICC) was measured using the method of Carter [44], with some modifications. Five different concentrations (6.25, 5, 3.13, 2.5, and 1.56 mg/mL) of the methanol extract and fractions were diluted in distilled water, 1 mM ferrozine (4,4-disulfonic acid sodium salt) solution, and 1 mM ferrous sulphate solution (*n* = 4). Algal samples (100 µL), FeSo_4_ solution (20 µL), and distilled water (40 µL) were mixed, and absorbance before adding ferrozine solution was taken at 562 nm. Ferrozine solution (40 µL) was mixed and incubated for 10 min at room temperature. A plate reading was recorded at 562 nm and the results were expressed as mg EDTA equivalent per 1 g of dry weight of the extract or fraction.

#### 4.5.5. Oxygen Radical Absorbance Capacity (ORAC)

The ORAC was measured using the method of Ou et al. [45], with modifications. Different concentrations (5, 2.5, and 1.25 mg/mL) of the extract and fractions were diluted in phosphate buffer saline (75 mM, pH 7.4). Before conducting the experiment, flourcein (16 mg in 100 mL of phosphate buffer saline) and AAPH [2,2′-azobis (2-amidinopropane) dihydrochloride] solutions (40 mg in 1 mL of phosphate buffer saline) were prepared. The reaction mixture (200 µL), containing algal samples (10 µL), PBS (40 µL) and fluorescein solution (100 µL), was preincubated at 37 °C for 5 min. After incubation, AAPH solution was added and absorbance recordings were made at excitation and emission wavelengths of 494 nm and 535 nm, respectively, at 1 min intervals for 35 min. Trolox was used as the standard, and the results were expressed as mg Trolox equivalent per 1 g dry weight of the sample.

### 4.6. In Vitro α-amylase and α-glucosidase Inhibitory Assay

#### 4.6.1. Alpha-Amylase Inhibitory Activity

The anti-amylase activity was measured using the method of Bernfeld [46], with modifications (*n* = 4). Various concentrations of the algal extract (400, 200, 100, 50, 25, and 12.5 µg/mL) and acarbose (200, 100, 50, 25, 12.5, and 6.25 µg/mL) were preincubated with starch (40 µL) and sodium acetate buffer (710 µL) at 40 °C in a shaking water bath (Wise Bath, WSB-30, Wertherim, Germany) for 10 min. The α-amylase enzyme solution (50 µl:1 mg/mL) was added and incubated again for 15 min. After incubation, 500 µL of 3,5-dinitrosalicylic acid (DNS) solution was added and boiled for 5 min until color developed and was then kept in an ice bath to cool. The absorbance was recorded at 540 nm using a SpectraMax Plus 384 instrument (Molecular Devices, San Jose, CA, USA). Another experiment was carried out in an identical way by replacing enzyme solution with acetate buffer to determine the absorbance produced by the sample itself. A control experiment was conducted by replacing extracts with 100 mM sodium acetate buffer. Acarbose was used as the standard and percentage α-amylase inhibition was calculated.

#### 4.6.2. Alpha-Glucosidase Inhibitory Activity

The α-glucosidase inhibition assay was performed according to the method of Matsui et al. [47], with some modifications. Different concentrations of algal samples (133.3, 66.6, 33.3, 16.6, 8.3, and 4.16 µg/mL) or acarbose (2.5, 1.25, 0.625, 0.313, 0.156, 0.078, and 0.039 µg/mL) in acetate buffer, and *p*-nitrophenyl-a-d-glucopyranoside (PNPG) solution were incubated at 37 °C for 5 min. After measuring the absorbance, 25 mU/mL of α-glucosidase was added and incubated at 37 °C for 35 min. After incubation, 10% NaCO_3_ solution was added to stop the reaction. The absorbance was then recorded at 400 nm. The reaction mixture without extract was used as the control and anti-glucosidase activity (% inhibition) was calculated.

#### 4.6.3. Glucose Diffusion Inhibitory Activity

Glucose diffusion inhibitory activity was determined using a dialysis bag and a glucose oxidase kit [48]. Initially, a dialysis membrane (D9777) was activated according to the manufacturer’s instructions. Briefly, 2 mL of the algae sample (1 mg/mL) was placed in a dialysis tube with 2 mL of 0.22 mM glucose solution. Subsequently, the dialysis tube was dipped in a beaker containing 80 mL of 0.15M NaCl and 20 mL of distilled water and was shaken constantly at 150 rpm at 37 °C. The glucose concentration (µg/mL) in the external solution was measured at 30 min intervals for 3 h using a GAGO-20 glucose oxidase kit (Sigma Aldrich). Acarbose and distilled water were used as the standard and the control, respectively.

#### 4.6.4. Antiglycation Activity

Antiglycation activity was measured using the method of Matsuura et al. [49], with some modifications. Algal extracts were prepared by dissolving with phosphate buffer saline. Bovine serum albumin (80 µL), algae extract (80 µL), 400 mM glucose (360 µL), and 50 mM PBS (pH 7.4, 480 µL) was incubated at 60 °C for 50 h. After incubation, the solution was cooled and 200 µl of 50% trichloroacetic acid was added and centrifuged at 15,000 rpm at 4 °C for 4 min. The resulting precipitate was dissolved in PBS (pH 10) and absorbance values were recorded at an excitation wavelength of 370 nm and emission wavelength of 440 nm (Amino-Bowman Series 2, Thermo Spectronic, Fitchburd, WI, USA). A sample negative was carried out in the same way, however without adding glucose. A sample control was conducted by replacing algal extracts with 50 mM PBS. Rutin was used as the standard and antiglycation activity (% inhibition) was calculated.

### 4.7. GC-MS Analysis of Extract and Solvent Fractions

The GC-MS analysis was performed on active fraction (ethyl acetate fraction) using a 5975C gas chromatograph (Agilent Technologies, Palo Alto, CA, USA) and an HP-5MS capillary column (30 m × 25 µm with a film thickness of 0.25 µm). Briefly, the sample (1 µL) was injected into the HP-5MS capillary column and was exposed to a temperature of 70 °C for 2 min; then, the temperature was increased from 70 °C to 200 °C at a rate of 3 °C/min^−1^ and held at 200 °C for 15 min. Helium was used as the carrier gas with a flow rate of 1 mL/min. After obtaining the chromatogram, the mass spectrum of the unknown component was identified using the NIST (NIST 17) library [50].

### 4.8. Statistical Analysis

Statistical analysis of each experiment was carried out using the Minitab 17 software (Cubic Computing Pvt. Ltd., Bangalore, India) and Microsoft Excel 2016. All the experiments were performed using four replicates. Mean and standard deviation were calculated using standard equations. One-way ANOVA was used to determine the significant differences between the methanol extract and the fractions of *G. edulis*. *p*-values less than 0.05 were considered significant. The Pearson’s correlation coefficient was used for the correlation analysis.

## 5. Conclusions

The present study found that *G. edulis* exhibited promising hypoglycemic activity by inhibiting key carbohydrate-digesting enzymes, glucose absorption, and the formation of antiglycation end products. Although antioxidant activity and enzyme inhibitory activity are stronger in commercial drugs than in *G. edulis* methanol extracts and its fractions, the hypoglycemic potential of *G. edulis* was evident in the present study. GC-MS analysis further confirmed the presence of bioactive compounds rich in antioxidants and antidiabetic properties. The ethyl acetate fraction exhibited promising antiglycation, and hypoglycemic potential. Hence, 3T3-L1 mature adipocytes and animal models are required to confirm anti-obesity and antidiabetic potentials of *G. edulis*. Isolation of active compounds for the development of new drugs is also warranted.

## Figures and Tables

**Figure 1 molecules-24-03708-f001:**
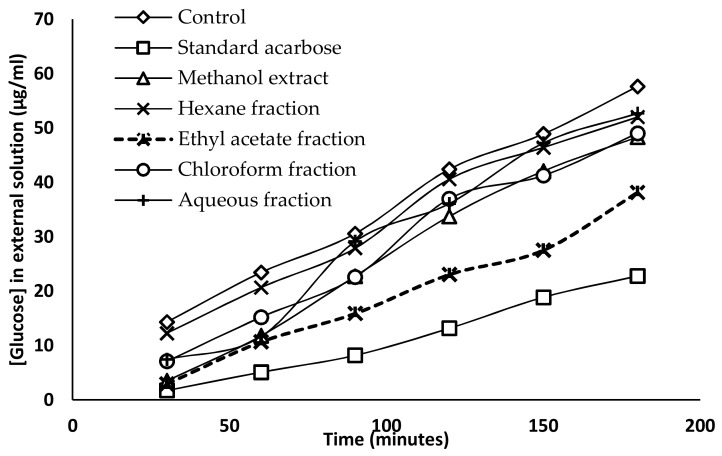
Effect of methanol extract and fractions of *G. edulis* (1000 µg/mL) on glucose diffusion through dialysis membrane compared to the standard acarbose and control. Data presented as means ± standard deviation (*n* = 4).

**Figure 2 molecules-24-03708-f002:**
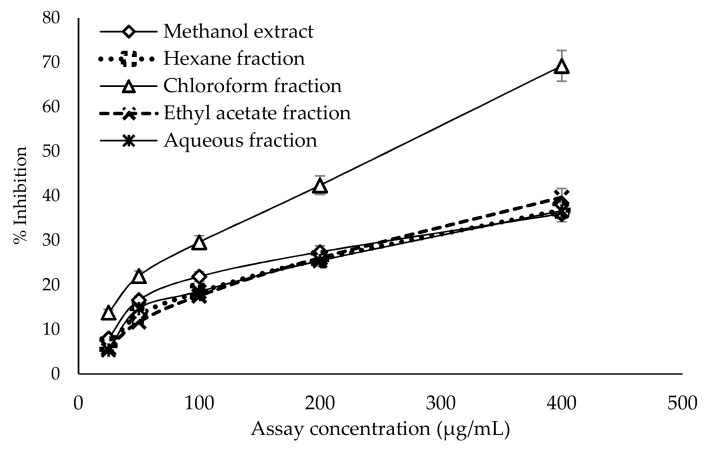
Dose–response relationship of methanol extract and its fractions of *G. edulis* for antiglycation activity determined by glucose-induced protein glycation and formation of protein-bound fluorescent advanced glycation end products. Data presented as mean ± standard deviation (*n* = 4).

**Figure 3 molecules-24-03708-f003:**
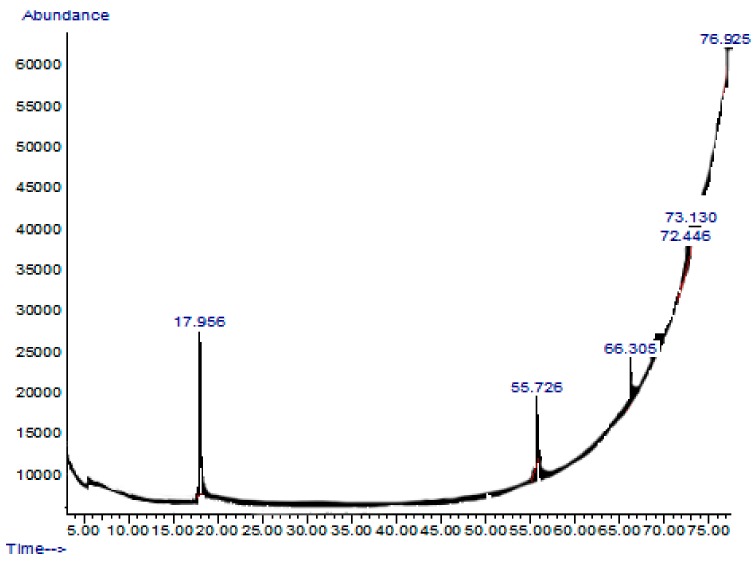
Chromatograms obtained from the gas chromatography-mass spectrometry (GC-MS) analysis of the ethyl acetate fraction of *G. edulis.*

**Table 1 molecules-24-03708-t001:** Phenol, flavonoid, and alkaloid contents of crude methanol extract and fractions of *Gracillaria edulis.*

Extract/Fraction	TPC (µg GAE/g)	TFC (µg QE/g)	Total Alkaloids (µg PE/g)
Crude methanol extract	1007.81 ± 54.21 ^a^	541.02 ± 51.84 ^a^	7177.72 ± 63.04 ^a^
Hexane fraction	760.85 ± 37.75 ^b^	688.60 ± 9.55 ^a^	1656.97 ± 45.80 ^b^
Chloroform fraction	560.85 ± 55.08 ^c^	289.39 ± 9.55 ^b^	2875.54 ± 22.29 ^c^
Ethyl acetate fraction	2414.51 ± 50.34 ^d^	1461.49 ± 75.22 ^c^	1073.75 ± 45.88 ^b^
Aqueous fraction	1704.69 ± 43.16 ^e^	786.95 ± 62.04 ^d^	522.34 ± 67.13 ^d^

TPC: total phenol content; TFC: total flavonoid content; GAE: gallic acid equivalent; QE: quercetin equivalent; PE: Piperine equivalent. Data presented as mean ± standard deviation (*n* = 4). Mean values in a column superscripted by different letters (^a–e^) are significantly different at *p* < 0.05.

**Table 2 molecules-24-03708-t002:** IC_50_ values of methanol extract of *G. edulis* and fractions against antioxidant activity and activities equivalent to standards.

Extract/ Fraction	IC_50_ (mg/mL)			Activity Equivalent to Standard (mg TE/g)	
	DPPH	ABTS	FICA	FRAP	ORAC
Crude methanol extract	3.19 ± 0.02 ^a^	0.56 ± 0.01 ^a^	9.23 ± 0.19 ^a^	0.26 ± 0.03 ^a^	1.61 ± 0.19 ^a^
Hexane fraction	6.22 ± 0.01 ^b^	0.54 ± 0.01 ^b^	2.58 ± 0.03 ^bc^	1.93 ± 0.35 ^b^	0.57 ± 0.07 ^bc^
Chloroform fraction	3.29 ± 0.02 ^c^	0.44 ± 0.01 ^c^	2.43 ± 0.01 ^c^	2.19 ± 0.23 ^b^	0.77 ± 0.05 ^b^
Ethyl acetate fraction	3.17 ± 0.04 ^a^	0.41 ± 0.02 ^d^	2.22 ± 0.01 ^bc^	8.51 ± 0.09 ^c^	1.44 ± 0.29 ^a^
Aqueous fraction	3.91 ± 0.03 ^d^	0.45 ± 0.03 ^c^	2.71 ± 0.02 ^b^	1.23 ± 0.21 ^d^	0.44 ± 0.09 ^c^
Trolox (standard)	0.011 ± 0.00 ^e^	0.008 ± 0.00 ^e^	-	-	-
EDTA (standard)	-	-	0.019 ± 00 ^d^	-	-

Results are expressed as mean ± SD; *n* = 4. DPPH (1,1-diphenyl-2-picrylhydrazine); ABTS (2,2′-azino-bis (3-ethylbenzothiazoline-6-sulphonic acid)); FICA (ferrous iron chelating activity); FRAP (ferric reducing antioxidant powder); ORAC (oxygen radical absorbance capacity); EDTA (ethylenediaminetetraacetic acid); TE (Trolox equivalent). Mean values in a column superscripted by different letters (^a–e^) are significantly different at *p* < 0.05.

**Table 3 molecules-24-03708-t003:** IC_50_ values exhibited by *G. edulis* methanol extract and methanol fractions against the inhibitory activity of the enzymes α-amylase and α-glucosidase and antiglycation activities.

Extract/Fraction	Alpha-Amylase (µg/mL)	Alpha-Glucosidase (µg/mL)	Anti-Glycation (µg/mL)
Crude methanol extract	349.59 ± 2.44 ^a^	102.24 ± 0.89 ^a^	702.33 ± 12.72 ^a^
Hexane fraction	393.04 ± 4.73 ^b^	163.90 ± 5.23 ^b^	637.53 ± 6.21 ^b^
Chloroform fraction	322.71 ± 4.80 ^c^	122.65 ± 2.37 ^c^	258.23 ± 3.24 ^c^
Ethyl acetate fraction	279.48 ± 5.62 ^d^	87.92 ± 1.62 ^d^	586.54 ± 4.37 ^b^
Aqueous fraction	376.49 ± 12.14 ^e^	148.57 ± 1.87 ^e^	723.78 ± 12.81 ^d^
Acarbose (standard)	87.43 ± 0.59 ^f^	0.38 ± 0.06 ^f^	-
Rutin (standard)	-	-	11.55 ± 0.82 ^e^

Results are expressed as mean ± SD (*n* = 4). **p* < 0.05 compared with the respective standard. Mean values in a column superscripted by different letters (^a–f^) are significantly different at *p* < 0.05.

**Table 4 molecules-24-03708-t004:** Active compounds identified in the ethyl acetate fraction of *G. edulis* by gas chromatography-mass spectrometry (GC-MS) analysis.

Retention Time and %	Name	Molecular Formula	Compound Class	Reported Biological Activity
17.956 (54.27%)	2,5-Dimethylhexane-2,5-dihydroperoxide	C_8_H_18_O_4_	Organic compound	Anti-inflammatory Antioxidant [14,15]
55.726 (11.46%)	Phthalic acid-6-ethyloct-3-yl 2-ethylhexyl ester	C_26_H_42_O_4_	Phthalic acid derivative	Anticancer Antimicrobial [16]
66.305 (15.39%)	1*H*-Indole-2-carboxylic acid,6-(4-ethoxyphenyl)-3-methyl-4-oxo-4,5,6,7-tetrahydro-isopropyl ester	C_19_H_20_FNO_3_	Indole derivative	Antidiabetic [17,18]
72.446 (10.93%)	1,2-dimethoxy-4 (1,3-dimethoxy-1-propenyl) benzene	C_13_H_18_O_4_	Benzene derivatives	Antifungal [19]
73.130 (1.52%)	Benz(b)1,4-oxazepine-4 (5*H*)-thione, 2,3-dihydro-2,8-dimethyl	C_11_H_13_NOS	Benzoxazepine derivatives	Anti-inflammatory Antimicrobial [20]
76.925 (6.49%)	2-acetoxymethyl-3-(methoxycarbonyl)biphenylene	C_17_H_14_O_4_	Biphenyl derivatives	Antibacterial [21]

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
