# Peer review of "In-Vitro Antioxidant, Hypoglycemic Activity, and Identification of Bioactive Compounds in Phenol-Rich Extract from the Marine Red Algae Gracilaria edulis (Gmelin) Silva"

_molecules, 2019, doi:10.3390/molecules24203708_

Round 1

Reviewer 1 Report

In this manuscript, entitled “Identification of bioactive compounds in phenolic rich extract from marine red algae Gracillaria edulis (Gmelin) Silva to combat obesity and type-2 diabetes” by Gunathilaka et al., the authors investigated the anti-oxidant and anti-diabetic effects of methanol extract and its fractions from red algae Gracillaria edulis (G. edulis) in-vitro. They found the ethyl acetate fraction might be the most potential source in all fractions and they identified six compounds from this fraction using GC-MS. Overall, this study is fine and worthy of publication in this journal. However, here are some questions or suggestions for the authors in order to validate their findings:

1. Since all the experiments were in vitro and it is difficult to find any experimental design related to obesity, the title of the manuscript is very inappropriate.

2. The authors did find 6 compounds in EA fraction showed in table 4, however, are they all phenolic compounds? Why did the authors emphasize “bioactive compounds in phenolic rich extract”?

3. The authors mentioned that the inhibitory activity of acarbose or the reference drug was considered as 100% for glucose diffusion inhibitory activity assay (lines 160-161, p. 5). However, the unit the authors used is glucose concentration (μg/mL).

4. The authors mentioned that amongst the tested fractions, ethyl acetate fractions exhibited maximum inhibition of glucose diffusion at 180 min and glucose concentration of the external solution was found to be 38.15±1.11μg/mL (lines 162-164, p. 5). However, the fraction showed highest inhibition of glucose diffusion at 180 min is chloroform fraction showed in figure 3.

5. The authors indicated that the highest DPPH radical scavenging activity was observed in ethyl acetate fraction (IC50: 3.19±0.02 mg/mL) and crude methanol extract (IC50: 3.19±0.02 mg/mL) compared to the standard Trolox (IC50: 0.011 mg/mL). However, the IC50 of EA fraction showed in table 2 is 3.17±0.04.

Author Response

The PDF file is attached 

Reviewer 2 Report

In this research article entitled “Identification of bioactive compounds in phenolic rich extract from marine red algae Gracillaria edulis (Gmelin) Silva to combat obesity and

Type-2 diabetes” the authors investigated the antidiabetic and anti-obesity potential of several extracts/fractions of the marine red algae Gracilaria edulis. They observed that fractions of the algae extract showed different degrees of antioxidant, iron chelating, and radical scavenging activities. Moreover, the ethyl acetate fraction, the most promising one, showed potent α-amylase and α-glucosidase inhibitory activity. This is an interesting study that may contribute to the development of new oral therapies based on extracts of Gracilaria edulis. However, this reviewer has several concerns that will be pointed out below.

General comments – major concerns

1) Language and general editing: although I did not have major issues reading the manuscript, I guess it would benefit from careful editing. Thus, I suggest the authors get editing help from someone with full professional/editorial proficiency in English.

2) Description of the results should be improved: sometimes the description of the results is redundant (e.g. Page 4, lines 135-142), and the authors mention the same results twice in the same paragraph. This could be improved by rewriting the paragraph to give the information only once in a concise and comprehensive way.

3) Chosen references and wrong use of the literature: this is a very important topic, which will be further explored below:

- the chosen reference does not support data: I could not understand why/how the authors have chosen most of the cited studies in this manuscript, as almost all of them simply do not support the data they are supposed to. For instance:
a) Page 2, References 5 (line 44), 4 (line 47), 1 (line 49), 6 (line 52), 7 (line 61), among many others (throughout the manuscript), were used to support data but these studies/reviews did not discuss/mention such point/data. That is, they were wrongly placed in the present manuscript;

b) Page 2, lines 68-73; reference 10: “Gracillaria edulis (Gmelin) Silva is a red algae… against LPS-induced inflammatory responses” – First, Shanura et al. did not measure antidiabetic, antimicrobial, anti-coagulant, anti-inflammatory and anti-proliferative activities in extracts of G. edulis as suggested by the authors of the present manuscript. In addition, Shanura et al. did not measure anti-inflammatory activity of methanol extract and fractions of G. edulis against LPS-induced inflammatory responses. Therefore, this Ref. 10 should not be included here.

4) IC50 calculation: As I am not a specialist in pharmacology, I would like to understand how the authors can be sure of the IC50 values calculated if their dose-response curves do not reach a plateau showing the extracts/fractions maximum effect. They did not reach a concentration in which a further inhibition is not observed (we can clearly see that the maximum concentration tested did not inhibit 100%, regardless of the measurement). Please look at the correct way to calculate IC50. If necessary, the authors need to complete their experiments with lower or higher concentrations of the extracts/fractions.

5) Effects in cellular models in vitro: The authors only show the inhibitory effects of G. edulis extracts/fractions in an isolated system using purified α-amylase and α-glucosidase. I would like to know whether they have tested this compound in a cellular, in vitro system. If not, I suggest the authors perform such experiments, as I believe it would increase the quality of the manuscript. Of note, this is only a suggestion, not a requirement for the acceptance of the manuscript.

6) Lack of appropriate experiments to determine the antidiabetic and anti-obesity effects of the G. edulis extracts/fractions: The authors state that their aim is “to appraise the antidiabetic and anti-obesity potential through inhibitory activities of carbohydrate digestive enzymes, glucose diffusion, and protein glycation.” However, I do not believe the authors managed to test their aim using the models described in the present manuscript.

- Antidiabetic effect

Herein, the authors measured carbohydrate digestive enzymes, glucose diffusion, and protein glycation. Although I agree that inhibition of α-amylase and α-glucosidase, as well as formation of AGE, are commonly used as parameters of antidiabetic properties in vitro, here the authors only measured the activity of these proteins in isolated systems (see point 5 above). In addition, I do not think that a glucose diffusion assay determined using dialysis tubes can really reflect the complexity of the intestinal glucose transport into the bloodstream.

- Anti-obesity effect

Regarding the anti-obesity effects, the authors should have performed, at least, some experiments in differentiated adipocytes (e.g. 3T3-L1 cell line), showing that their extracts/fractions have antiadipogenic effects (e.g. inhibition of lipid accumulation).

- In vivo studies

Finally, I believe it is very difficult to discuss antidiabetic and anti-obesity effects without experimentation in vivo. I understand in vivo experiments are not precisely easy to address, but at least two simple experiments showing fasting glucose levels (experiment 1) and an oral glucose tolerance test (experiment 2) upon treatment with the adequate vehicles and G. edulis extracts/fractions need to be performed.

Specific comments (divided by sections of the manuscript)

Title

1) The correct name of the algae is Gracilaria edulis, with only one “l” in “Gracilaria”.

Introduction

1) Page 1, line 35: Please use an official reference to address this topic, such as Ref. 2. The authors can also consult the International Diabetes Federation atlas for more information (https://www.diabetesatlas.org/).

2) Page 2, line 77: “(…) samples were collected from different regions of the world (…)” – I do not fully agree with this statement. Based on the three studies presented so far in the manuscript, one collected samples from Sri Lanka (Shanura et al. 2017), while the other two (Koneri et al. 2017 and Patra et al. 2013) collected samples in the Mandapam region, India. As this region is geographically very near to Kalpitiya, Sri Lanka – where the authors of the present manuscript collected their samples –, I think it is an overstatement to say that other studies collected their samples from different regions of the world.

Results

3) Page 3, lines 97-102: “The total flavonoid content… obtained in the chloroform fraction.” – This can be summarized in fewer words. It is very confused in the present form.

4) Page 3, lines 111-112: “Free radical scavenging activity… scavenging activities.” – It is repetitive. The same has been stated in lines 107-108.

5) Page 3, line 113: The value for IC50 presented in the text (i.e. 3.19±0.02 mg/mL) do not correspond to what is presented in Table 2 (i.e. 3.17±0.04 mg/mL). Please change it accordingly.

6) Page 4, lines 135-142: “The ethyl acetate fraction… fraction (Table 3).” – This can be summarized in fewer words. It is very redundant in the present form.

7) Page 5, lines 148-150: “The α-glucosidase activity… presented in Table 3.” – It is not the α-glucosidase activity of the crude methanol extract and the fractions, but the inhibitory activity of such extract and fractions that are being shown. This sentence should be rewritten.

8) Page 5, lines 164-166: “The inhibition… more or less similar (Table A1).” – This “more or less similar” is compared to what? This needs clarification.

9) Page 6, lines 183-185: “The crude methanol extract… inhibitory activities.” – This is not true. Only the chloroform fraction showed a different degree of inhibition, whereas all other fractions showed similar inhibitory activity.

10) Page 8, Ref. 13: I could not find the compound “1H-Indole-2-carboxylic acid,6-(4-ethoxyphenyl)-224 3-methyl-4-oxo-4,5,6,7-tetrahydro-isopropyl ester” in the cited reference. Is there any other publication to support this statement, i.e. that this compound presents antidiabetic activities? Please cite other studies to support this sentence.

Discussion

11) Page 8, lines 244-247: “Oxidative stress is linked… production of adipocytokines.” – This is much more complex than simply activation of NADPH oxidase and decreased adipocytokine production. I would like to suggest the inclusion of a Review article as a reference for this subject, as it would be more appropriate to cover this wide subject.

12) Page 8, lines 244-247: “Therefore, the supplementation of antioxidants can reduce the risk of being obese.” – Abdali D and colleagues (Med Princ Pract. 2015;24(3):201-15) state that “the literature does not suggest antioxidant supplementation as a cure-all for obesity or for type 2 diabetes”, which contrasts with the statement made by the authors of the present manuscript. Therefore, the authors must include studies supporting the statement made in the manuscript.

13) Page 8, line 262: “(…) ethyl acetate fraction (0.43 and 0.809 mmol TE/g and).” – It seems that a piece of information is missing in this part of the sentence between parenthesis.

14) Page 8, line 264: “(…) the highest ferrous iron chelating activity (FICA) (…)” – Why did the authors change from “FICC” to “FICA”? This should be clarified and/or corrected if necessary.

15) Page 9, lines 272-273: “Inhibition of key metabolic carbohydrate digestive enzymes… medicinal plants.” – The authors must include studies supporting the statement made in the manuscript.

16) Page 9, lines 283-284: “In the present study… previous study used water as a solvent.” – Which is this “previous study” mentioned by the authors?

17) Page 9, lines 286-287: “Then the glucose is diffused through the intestinal wall and increases the postprandial blood glucose level.” – This mechanism is not so simple and relies on several enzymes (e.g. SGLT1 and Na+/K+-ATPase) and transporters (GLUT2) that will facilitate glucose absorption by intestinal cells and its transport into the bloodstream. The authors should be careful with this sort of statement to not give the wrong information to readers.

18) Page 9, lines 294-296: “Therefore, the finding of this assay… inhibiting glucose absorption.” – I disagree with this statement because, as explained above (point 17), glucose absorption by intestinal cells is not a simple mechanism depending only on diffusion through a membrane.

19) Pages 9-10, lines 307-309: “Further, GC-MS analysis… 2,5-dimethylhexane-2,5-309 dihydroperoxide.” – There is not enough evidence in the literature to support this statement. Once again, the authors should be careful with this sort of statement.

Materials and Methods

20) Page 10, lines 314-315: The title of the topic is repeated.

21) Page 11, line 347: “(…) pre-plate reading was taken at 415nm.” – What does “pre-plate reading” mean? This should be clarified, as it is used throughout the methods’ section.

22) Page 12, line 412: “The anti-amylase activity (…)” – I believe the authors mean “The a-amylase activity”.

23) Page 12, line 419: “Sample negative was carried out in an identical way without adding enzyme.” – This is not a “sample negative”, but an “enzyme negative” control.

24) Page 14, line 462: “All the experiments were performed using four replicates.” – Are these four replicates from four different algae extractions or only experiments repeated four times with the same algae sample? In other words, did the authors test different algae extractions in their study?

25) Regarding Statistical analysis in general: the authors mentioned they performed One-way ANOVA as statistical test, but only showed statistics in Table 1 (or at least they included different letters among different fractions). The authors must include statistical analyses in all tables/figures and show comparisons among fractions.

References

26) Formatting: the references’ format is not following the journal’s guidelines and, then, must be formatted accordingly.

27) References mentioned in the text that do not discuss/support the statement they are supposed to: Refs 1, 5, 4, 6, 7, 9, 10, 13, 14, 17, 19, 21, 25, 26, and 27.

28) Reference 15: the name of the journal is wrong. The correct one is “Oxid Med Cell Longev.”. Please change it accordingly.

29) Reference 18 has been previously cited as Ref. 7. Please change it accordingly.

30) Reference 29 does not describe the technique the authors say it describes. Please change it accordingly.

Figures and Tables

31) I did not understand why the authors separated in “Figures A” or “Figure 3”? Figures A1, A2 and A3 are very important to be considered as “Suppl. Material”. In addition, “Figure 1” and “Figure 2” are missing in the main manuscript.

Furthermore, figure legends must be improved and give a better description of the figure as well as statistical analysis where appropriate.

32) In Table 1, what does the letter a-e mean? It is not written in Table legend.

33) In Table 4, the authors must provide the literature supporting each one of the “reported biological activities” mentioned in the table.

Reviewer 3 Report

The manuscript entitled “Identification of bioactive compounds in phenolic rich extract from marine red algae Gracillaria edulis (Gmelin) Silva to combat obesity and Type-2 diabetes.” examines about inhibitory effects of several fractions obtained from marine red algae Gracillaria edulis against diabetes related parameters in vitro. The IC50 values of the fractions were determined and compared with positive control samples. Moreover, compositions of the fractions were determined and the data suggest that the fractions contain effective compounds.

This study includes beneficial information for developing new anti-diabetes fractions from the marine sample.

Comments:

1). In Table 3, IC50 value of methanol extract for anti-glycation was higher than those of hexane fraction, chloroform fraction, and ethyl acetate fraction. On the other hand, in Figure 4, inhibitory activity of methanol extract was higher than other fractions. These results are apparently opposite. How do the authors explain?

2). The IC50 values of fractions against alpha-amylase, alpha-glucosidase, and anti-glycation, were one or two-order higher than positive controls. The result suggests that the inhibitory activities are markedly low. Therefore, it is difficult to explain the anti-diabetic activity of the marine sample by the inhibitory activities of the parameters. The authors should explain the relationship between the effective dose of marine sample and the inhibitory dose of fractions in discussion section.

3). The English writing of the manuscript should be improved. The authors should check again the English in the manuscript.

Round 2

Reviewer 2 Report

First, I would like to say that the authors have done a commendable job in their revisions to this manuscript: they have edited the language, rewritten some parts of the text and changed most of the bibliography that was not correct in their first submission.

The manuscript has significantly improved, but I still have 2 major concerns that have not been properly addressed.

1) Effects in cellular models in vitro and lack of appropriate experiments to determine the antidiabetic and anti-obesity effects of the G. edulis extracts/fractions:

The authors claimed they used four main experiments to determine the anti-diabetic effect of G. edulis, namely inhibition of α-amylase and α-glucosidase, formation of AGEs, and glucose diffusion through dialysis membranes. However, I still think that it is an overstatement to attribute “antidiabetic and anti-obesity effects” based only on these set of experiments as type 2 diabetes and obesity are two very complex diseases. Moreover, I do not see how the activities of α-amylase and α-glucosidase are used “to determine the anti-obesity effect” of the extracts/fractions.

Without further experiments in cell and/or animal models mimicking these diseases, it is very difficult to attribute “antidiabetic and anti-obesity effects” to these G. edulis extracts/fractions. At most, the authors can say that these extracts/fractions are able to inhibit α-amylase and α-glucosidase as well as decrease formation of AGEs, and, thus, may be promising therapeutic agents in the combat of type 2 diabetes.

Hence, in my opinion, the authors should change the focus of their manuscript, because the title is somewhat misleading, as a type 2 diabetes model is not assessed herein.

2) Page 15, line 2097 in the revised version: “All the experiments were performed using four replicates.” – Are these four replicates from four different algae extractions or only experiments repeated four times with the same algae sample? In other words, did the authors test different algae extractions in their study?

Authors’ response: It means the same experiment repeated four times to reduce errors.

If the authors simply repeated the same experiments four times with the same algae extracts/fractions, how can they be sure that a new, different algae extraction will give the same results? The authors need to repeat the experiments with different plant extracts to provide evidence that different algae samples extracted in different days will give the same – or very similar – results. I understand there might be variation in the content of phenols, flavonoids, and alkaloids, but the effects on α-amylase and α-glucosidase, and formation of AGEs must be similar to the ones described in the present manuscript. Otherwise, it will be difficult to state these algae extracts/fractions may be used as therapeutic adjuvants in the treatment of type 2 diabetes.

Author Response

 First, I would like to say that the authors have done a commendable job in their revisions to this manuscript: they have edited the language, rewritten some parts of the text and changed most of the bibliography that was not correct in their first submission.
The manuscript has significantly improved, but I still have 2 major concerns that have not been properly addressed.

Point 1. Effects in cellular models in vitro and lack of appropriate experiments to determine the antidiabetic and anti-obesity effects of the G. edulis extracts/fractions:The authors claimed they used four main experiments to determine the anti-diabetic effect of G. edulis, namely inhibition of α-amylase and α-glucosidase, formation of AGEs, and glucose diffusion through dialysis membranes. However, I still think that it is an overstatement to attribute “antidiabetic and anti-obesity effects” based only on these set of experiments as type 2 diabetes and obesity are two very complex diseases. Moreover, I do not see how the activities of α-amylase and α-glucosidase are used “to determine the anti-obesity effect” of the extracts/fractions.

Without further experiments in cell and/or animal models mimicking these diseases, it is very difficult to attribute “antidiabetic and anti-obesity effects” to these G. edulis extracts/fractions. At most, the authors can say that these extracts/fractions are able to inhibit α-amylase and α-glucosidase as well as decrease formation of AGEs, and, thus, may be promising therapeutic agents in the combat of type 2 diabetes. Hence, in my opinion, the authors should change the focus of their manuscript, because the title is somewhat misleading, as a type 2 diabetes model is not assessed herein.

Thank you the fruitful comment which helped to improve our manuscript immensely. We have already exhausted our grant and Sri Lanka being a developing country, it is difficult to obtain further funding for the research. Therefore, at the moment it is difficult for us to complete the cellular models. As suggested by the reviewer, we revised the title of the manuscript as “In-vitro antioxidant, hypoglycemic activities and identification of bioactive compounds in phenol-rich extract from the marine red algae Gracilaria edulis (Gmelin) Silva”. We have removed the anti-obesity part of the manuscript (title and content).

Point 2.  Page 15, line 2097 in the revised version: “All the experiments were performed using four replicates.” – Are these four replicates from four different algae extractions or only experiments repeated four times with the same algae sample? In other words, did the authors test different algae extractions in their study?

Authors’ response: It means the same experiment repeated four times to reduce errors.

If the authors simply repeated the same experiments four times with the same algae extracts/fractions, how can they be sure that a new, different algae extraction will give the same results? The authors need to repeat the experiments with different plant extracts to provide evidence that different algae samples extracted in different days will give the same – or very similar – results. I understand there might be variation in the content of phenols, flavonoids, and alkaloids, but the effects on α-amylase and α-glucosidase, and formation of AGEs must be similar to the ones described in the present manuscript. Otherwise, it will be difficult to state these algae extracts/fractions may be used as therapeutic adjuvants in the treatment of type 2 diabetes.

Thank you for your comment which improve our manuscript immensely. We have collected the G.edulis algae sample during the period of February and we were able to collect the sufficient amount of sample to conduct the assays. Before the extraction, the whole sample was washed, dried and ground into a fine powder and mixed together to homogenize the sample and took the necessary amount from the bulk of the homogenized sample for the extraction. According to the literature on G.edulis (Francavilla, 2013) and as reviewer mentioned, the total phenolic content and antioxidant activity of a particular algae sample was varied with the seasonal variations and climatic changes. In addition, literature showed that seasonal variations affect the anti-diabetic activities of medicinal plants (Kolawole and Ayankunle, 2012; Burm, 2017). Therefore, in the present study, we have taken the homogenized sample collected in a particular time period (February) to avoid the effects of seasonal variations and climatic changes. In addition, we have repeated the experiments four times in order to minimize the errors.

§  Francavilla, M. The red seaweed Gracilaria gracilisas a multi products source. Marine Drugs2013, 11(10), 3754–3776.
§  Kolawole, O. T. and Ayankunle, A. A. (2012) ‘Seasonal Variation in the Anti-Diabetic and Hypolipidemic Effects of Momordica charantia Fruit Extract in Rats’.European Journal of Medicinal Plants, 2(2), pp. 177–185.
§  Burm, L. (2017) ‘Effect of climate change on phytochemical diversity, total phenolic content and in vitro antioxidant activity of Aloe’, BMC Research Notes. BioMed Central, pp. 1–12. doi: 10.1186/s13104-017-2385-3.

Reviewer 3 Report

Gunathilaka TL et al performed the experiments about inhibitory effects of several fractions obtained from marine red algae Gracillaria edulis against diabetes related parameters in vitro and the IC50 values of the fractions were determined. Moreover, compositions of the fractions were determined by Gas chromatography-mass spectrometry and the fractions contain effective compounds. These results suggest that the marine samples have anti-diabetic related activities. The manuscript includes helpful information for developing new anti-diabetic drugs using the marine samples.

Comments:

1). The Figure legends of the Figures should be described in detail for understanding.

2). The Figure 3 is difficult to see. Therefore, the Figure 3 should be improved.

Author Response

Point 1. The Figure legends of the Figures should be described in detail for understanding.

 Thank you for the fruitful comments and we have changed the figures accordingly.

Point 2. The Figure 3 is difficult to see. Therefore, the Figure 3 should be improved.

Thank you for pointing out this and we have enlarged the graph accordingly.

Round 3

Reviewer 2 Report

The manuscript has significantly improved, but I still have 2 major concerns that have not been properly addressed.

1) Although the authors have not addressed this concern, they have slightly changed the manuscript in order to suggest that the extracts tested herein have therapeutic potential.

However, I am still concerned with the fact that the authors did not change the paragraph between lines 396 and 408 in the page 8 (revised manuscript). If they recognize that intestinal glucose transport into the blood stream depends on GLUT-2 transporter and Na+-K+ pump, why did they keep the glucose diffusion assay to mimic intestinal glucose transport? I do not think that a glucose diffusion assay determined using dialysis tubes can really reflect the complexity of the intestinal glucose transport into the blood stream, let alone conclude that the ethyl acetate fraction of G. edulis has a therapeutic potential to reduce postprandial blood glucose based on this assay.

2) I believe the authors did not understand my point regarding the number of experiments. My main concern is the fact that they used only one preparation and repeated their experiments four times using this same preparation instead of different ones.

As I said before, I know that the content of phenols, flavonoids, and alkaloids may change according to seasonal variations. My point is that the authors cannot assure that a different preparation from samples collected next February will give the same results, as they have only prepared one extraction and tested it four times. In fact, one would expect that four experiments performed with the same preparation would give similar results, which is the case in the present manuscript.

As I believe the authors will not wait until next February to collect new samples and perform the whole study again, I guess we will not reach an agreement here.

Author Response

1) Although the authors have not addressed this concern, they have slightly changed the manuscript in order to suggest that the extracts tested herein have therapeutic potential.

However, I am still concerned with the fact that the authors did not change the paragraph between lines 396 and 408 in the page 8 (revised manuscript). If they recognize that intestinal glucose transport into the blood stream depends on GLUT-2 transporter and Na+-K+ pump, why did they keep the glucose diffusion assay to mimic intestinal glucose transport? I do not think that a glucose diffusion assay determined using dialysis tubes can really reflect the complexity of the intestinal glucose transport into the blood stream, let alone conclude that the ethyl acetate fraction of G. edulishas a therapeutic potential to reduce postprandial blood glucose based on this assay.

Thank you very much for pointing out this fact again and as a reviewer suggested we understood that glucose diffusion inhibitory assay does not reflex the complexity of the small intestine. Therefore, the sentence has changed accordingly as follows.

The glucose diffusion inhibition test was carried out to evaluate the effect of methanol extract and fractions of G.eduliswith respect to its glucose retardation activity across the dialysis tube. The glucose entrapment ability of the crude methanol extract and four fractions were found to be significantly different at different times. Among them, the ethyl acetate fraction of G.edulisexhibited a significant glucose entrapment ability which decreased the glucose movement into the external solution at 180 minutes compared to the control. The fact that the ethyl acetate fraction exhibited the highest inhibition of glucose diffusion may be due to the presence of insoluble fiber particles which entrap glucose molecules [33]. Since, the dialysis tube method is a simple technique, which only determine the potential effect of methanol extract and fractions of G.edulis to retard the glucose diffusion through the normal dialysis membrane, whereas in the intestinal tract, transportation of glucose is assisted by glucose transporters incorporated with other molecules in addition to the intestinal contractions [33].Therefore, further in-vivo studies should carry out to determine the real effect of methanol extract and fractions of G. edulis on glucose diffusion.

2) I believe the authors did not understand my point regarding the number of experiments. My main concern is the fact that they used only one preparation and repeated their experiments four times using this same preparation instead of different ones.

As I said before, I know that the content of phenols, flavonoids, and alkaloids may change according to seasonal variations. My point is that the authors cannot assure that a different preparation from samples collected next February will give the same results, as they have only prepared one extraction and tested it four times. In fact, one would expect that four experiments performed with the same preparation would give similar results, which is the case in the present manuscript.

As I believe the authors will not wait until next February to collect new samples and perform the whole study again, I guess we will not reach an agreement here

Thank you very much for pointing out this and we agreed with the fact that the reviewer emphasized. During the sample collection, initially, we have collected our samples in February 2018 from the Kalpitiya area to conduct these experiments. According to the results of in-vitro hypoglycemic activity, we wanted to send the fractions of G.edulis to Canada to isolate the compounds related to hypoglycemic activity and still it is in processing. We have collected the algae sample again in February 2019 and carried out extractions and fractionation to send the fractions to Canada. During the procedure, they too determined the α-amylase and α-glucosidase inhibitory activity of fractions. They too obtained similar results for the α-amylase and α-glucosidase inhibitory activitywith regard to IC50values. Sri Lanka being a tropical country variations are minimum unlike temperate countries. We only have monsoon and inter monsoon seasons. The weather is more or less similar in a particular district. Country can be divided in to 3 zones; wet zone, dry zone and arid zone and the soil and weather conditions vary in these zones but within zones it is more or less similar except for the rain. I am so sorry for this disparity.

But as a reviewer suggested we cannot assure about the results of phytochemicals, antioxidants of methanol extract and fractions obtained from the particular sample collected in February 2019. But compared to the results of α-amylase and α-glucosidase inhibitory activities, we could think that, methanol extract and fractions obtained from particular sample may have similar content of phytochemicals as in the sample collected in February 2018. Further, we have repeated each experiment four times to ensure the reliability of the data we obtained. I am so sorry for this disagreement .
